# Challenges in Context-Aware Neural Machine Translation

**Linghao Jin**[†1] **Jacqueline He**[†2] **Jonathan May**[1] **Xuezhe Ma**[1]

[1]Information Sciences Institute, University of Southern California

[2]University of Washington

{linghaoj, jonmay, xuezhema}@isi.edu   jyyh@cs.washington.edu

## Abstract

Context-aware neural machine translation, a paradigm that involves leveraging information beyond sentence-level context to resolve inter-sentential discourse dependencies and improve document-level translation quality, has given rise to a number of recent techniques. However, despite well-reasoned intuitions, most context-aware translation models yield only modest improvements over sentence-level systems. In this work, we investigate and present several core challenges, relating to discourse phenomena, context usage, model architectures, and document-level evaluation, that impede progress within the field. To address these problems, we propose a more realistic setting for document-level translation, called paragraph-to-paragraph (PARA2PARA) translation, and collect a new dataset of Chinese-English novels to promote future research.[1]

## 1 Introduction

Neural machine translation (NMT) has garnered considerable scientific interest and commercial success in recent years, with current state-of-the-art systems approaching or exceeding human quality for a few resource-rich languages when translating individual sentences (Wu et al., 2016; Hassan et al., 2018; Yang et al., 2020). Despite the strong empirical performance of such systems, the independence assumption that underlies sentence-level NMT raises several issues. Certain textual elements, such as coreference (Guillou and Hardmeier, 2016), lexical cohesion (Carpuat, 2009), or lexical disambiguation (Rios Gonzales et al., 2017) are impossible to correctly translate without access to linguistic cues that exist beyond the present sentence (Sim Smith, 2017). When evaluating documents rather than individual sentences, the adequacy and fluency of professional human

translation continues to surpass that of MT systems (Läubli et al., 2018), thus underscoring the need for incorporating long-range context.

Despite some efforts to meaningfully exploit inter-sentential information, many context-aware (or interchangeably, document-level) NMT systems only show meager gains across sentence-level and document-level translation metrics (Tiedemann and Scherrer, 2017; Miculicich et al., 2018; Müller et al., 2018; Tu et al., 2018; Maruf et al., 2019; Lupo et al., 2022a,b; Wu et al., 2022). Performance improvements against sentence-level baselines on overall translation accuracy, pronoun resolution, or lexical cohesion become less pronounced when context-aware systems are trained on realistic, high-resourced settings (Lopes et al., 2020), casting doubt on the efficacy of such approaches.

In this paper, we conduct a thorough empirical analysis and present some key obstacles that hinder progress in this domain:

1. Existing document-level corpora contain a sparse number of discourse phenomena that require inter-sentential context to be accurately translated.

2. Though context is necessary for pronoun resolution and named entity consistency, it is less helpful for tense and discourse markers.

3. The sentence-level Transformer baseline already performs up to par with concatenation-based NMT settings.

4. Advanced model architectures do not meaningfully improve document-level translation on existing document-level datasets.

5. Current metrics designed for document-level translation evaluation do not adequately measure document-level translation quality.

The above findings suggest that paragraph-to-paragraph (PARA2PARA) translation, wherein a

---

†Equal contribution.

[1]We release the paper's code and dataset here: https://github.com/Linghao-Jin/canmt-challenges.

document is translated at the granularity of paragraphs, may serve as a more suitable and realistic setting for document-level translation, which in practice is unencumbered by sentence-level alignments. To this end, we develop and release a new paragraph-aligned Chinese-English dataset, consisting of 10,545 parallel paragraphs harvested from 6 novels within the public domain, in order to spur future research.

## 2   Background

The high-level objective of *sentence*-level machine translation is to model the sentence-level conditional probability $P(\boldsymbol{y}|\boldsymbol{x})$, in which the source and target sentences $\boldsymbol{x} = (x^1, ..., x^M)$, $\boldsymbol{y} = (y^1, ..., y^N)$ are textual sequences of respective lengths $M$ and $N$. Under the dominant paradigm of neural machine translation (Sutskever et al., 2014), the conditional probability $P_\theta(\boldsymbol{y}|\boldsymbol{x})$ is typically decomposed into the following auto-regressive formulation (with $\theta$ denoting parameterized weights):

$$P_\theta(\boldsymbol{y}|\boldsymbol{x}) = \prod_{n=1}^{N} P_\theta(y^n|\boldsymbol{x}, y^{<n}). \quad (1)$$

Equation 1 implies that when predicting the target token $y^n$, the model could only access the current source sentence, $\boldsymbol{x}$, as well as all previously translated tokens in the current target sentence, $y^{<n}$. Translating sentences in a document in such an isolated fashion, without any extra-sentential information that lies beyond sentence boundaries, has been found to produce syntactically valid, but semantically inconsistent text (Läubli et al., 2018).

To remedy this, *context-aware* neural machine translation considers a document $D$ that entails a set of logically cohesive source sentences $\boldsymbol{X} = \{\boldsymbol{x}_1, \boldsymbol{x}_2, ..., \boldsymbol{x}_d\}$, and a parallel set of target sentences $\boldsymbol{Y} = \{\boldsymbol{y}_1, \boldsymbol{y}_2, ..., \boldsymbol{y}_d\}$. Under a left-to-right translation schema, the model computes the probability of translating the source sentence $\boldsymbol{x}_i$ conditioned on the context $C_i$, wherein $0 \leq i \leq d$:

$$P_\theta(\boldsymbol{y}_i|\boldsymbol{x}_i, C_i) = \prod_{j=1}^{N} P_\theta(y_i^j|\boldsymbol{x}_i^j, y_i^{<j}, C_i). \quad (2)$$

In practice, there are multiple ways to formulate $C_i$. Passing in $C_i = \{\emptyset\}$ reduces to the sentence-level case (1). Throughout this paper, we explore two concatenation-based setups first presented by Tiedemann and Scherrer (2017). The **one-to-two** (1-2) setup prepends the preceding target sentence to the current target sentence ($C_i = \{y_{i-1}\}$), denoting sentence boundaries with a <SEP> token.

The **two-to-two** (2-2) setup incorporates additional context from the previous source sentence ($C_i = \{x_{i-1}, y_{i-1}\}$). The target context is integrated in the same manner as in **one-to-two**.

In order to investigate the importance of context after the current sentence, we also explore a **three-to-one** (3-1) setting, wherein we introduce additional source-side context by concatenating the previous and subsequent sentences to the current one ($C_i = \{x_{i-1}, x_{i+1}\}$), and do not incorporate any target context.

## 3   Related Work

### 3.1   Model Architectures

Recent progress in context-aware NMT generally falls along two lines: multi-encoder approaches and concatenation-based ones (Kim et al., 2019).

Under the first taxonomy, additional sentences are encoded separately, such that the model learns an internal representation of context sentences independently from the current sentence. The integration of context and current sentences can occur either prior to being fed into the decoder (Maruf and Haffari, 2018; Voita et al., 2018; Miculicich et al., 2018; Zhang et al., 2018; Maruf et al., 2019), or within the decoder itself (Bawden et al., 2018; Cao and Xiong, 2018; Kuang and Xiong, 2018; Stojanovski and Fraser, 2018; Tu et al., 2018; Zhang et al., 2018). The effectiveness of these multi-encoder paradigms is subject to debate; in a standardized analysis, Li et al. (2020) finds that rather than effectively harnessing inter-sentential information, the context encoder functions more as a noise generator that provides richer self-training signals, since even the inclusion of random contextual input can yield substantial translation improvement. In addition, Sun et al. (2022) finds that BLEU-score improvements from context-aware approaches often diminish with larger training datasets or thorough baseline tuning.

On the other hand, concatenation-based NMT approaches are conceptually simpler and have been found to perform on par with or better than multi-encoder systems (Lopes et al., 2020; Ma et al., 2021). Under this paradigm, context sentences are appended to the current sentence, with special tokens to mark sentence boundaries, and the concatenated sequence is passed as input through the encoder-decoder architecture (Ma et al., 2020).

## 3.2 Datasets

Until recently, the bulk of context-aware NMT research has focused on document-level, *sentence*-aligned parallel datasets. Most commonly used corpora, including IWSLT-17 (Cettolo et al., 2012), NewsCom (Tiedemann, 2012), Europarl (Koehn, 2005), and OpenSubtitles (Lison et al., 2018) are sourced from news articles or parliamentary proceedings. Such datasets often contain a high volume of sentences that is sufficient for training sentence-level NMT systems, yet the number of *documents* remains comparatively limited.[2]

In an attempt to address the scarcity of document-level training data, recent works have developed datasets that are specifically tailored for context-aware NMT. Jiang et al. (2023) curated Bilingual Web Books (BWB), a document-level parallel corpus consisting of 9.6 million sentences and 196 thousand documents (chapters) sourced from English translations of Chinese web novels. Thai et al. (2022) introduced PAR3, a multilingual dataset of non-English novels from the public domain, which is aligned at the paragraph level based on both human and automatic translations. Using automatic sentence alignments, Al Ghussin et al. (2023) extracted parallel paragraphs from Paracrawl (Bañón et al., 2020), which consists of crawled webpages.

## 3.3 Evaluation

In addition to metrics that evaluate sentence-level translation quality, e.g., BLEU (Papineni et al., 2002) and COMET (Rei et al., 2020), a number of automatic metrics designed specifically for document-level MT have been recently proposed. Jiang et al. (2022) introduced BlonDe, a document-level automatic metric that calculates the similarity-based F1 measure of discourse-related spans across four categories. Vernikos et al. (2022) show that pre-trained metrics, such as COMET, can be extended to incorporate context for document-level evaluation. To measure the influence of context usage in context-aware NMT models, Fernandes et al. (2021) proposed Context-aware Cross Mutual Information (CXMI), a language-agnostic indicator that draws from cross-mutual information.

Another approach to document-level MT evaluation focuses on hand-crafted contrastive evaluation sets to gauge the model's capacity for capturing inter-sentential discourse phenomena, including ContraPro (Müller et al., 2018) in English-to-German, Bawden (Bawden et al., 2018) in English-to-French, and Voita (Voita et al., 2019) in English-to-Russian translation. Though targeted, these test sets tend to be small, and are constricted to a particular language pair and discourse phenomenon.

## 4 Challenges

We identify key obstacles that account for the lack of progress in this field, based on a careful empirical analysis over a range of language pairs, model architectures, concatenation schemas, and document-level phenomena.[3]

### 4.1 Discourse phenomena is sparse in surrounding context.

*Contextual sparsity* is a bottleneck to document-level neural machine translation that manifests in two forms (Lupo et al., 2022a). First, the majority of words within a sentence can be accurately translated without additional access to inter-sentential information; context poses as a weak training signal and its presence has not been found to substantially boost translation performance. Second, only a few words in neighboring sentences may actually contribute to the disambiguation of current tokens at translation time.

We investigate contextual sparsity via a fine-grained analysis on the BWB (Jiang et al., 2022) test set, which has been manually tagged with specific discourse-level phenomena.[4] Specifically, we use it to probe NMT models' ability to exploit long-range context by analyzing the frequency of particular discourse phenomena that can only be resolved with context.

For the manual analysis, we randomly sample 200 discourse-annotated instances from the test set and ask bilingual annotators who are fluent in Chinese and English to identify and count instances that contain a particular context-dependent discourse phenomenon. Annotators are asked to discern if the following document-level discourse phenomena exist in each sentence pair:

- **Pronoun Ellipsis:** The pronoun is dropped in Chinese, but must be included in the English translation.

---

[2]As an example, the IWSLT-17 (Cettolo et al., 2012) EN→FR split contains 239854 sentences and 2556 documents.

[3]Unless otherwise specified, all experiments are conducted using Transformer (Vaswani et al., 2017) as the default architecture. Full training details are in Appendix A.

[4]Examples of annotated paragraphs are in Appendix A.4.

- **Lexical Cohesion:** The same named entity must be translated consistently across the current sentence and context sentences.

- **Tense:** Tense information that can be omitted in Chinese, and must be inferred based on context to be correctly translated in English.

- **Ambiguity:** Instances in which an ambiguous word or phrase in the current sentence requires context to be correctly translated.

- **Discourse Marker:** A discourse marker, e.g., *while*, *as long as*, *else*, that is not explicit in Chinese, but must be pragmatically inferred and present in English.[5]

Table 1 indicates that lexical cohesion (83.2%) and pronoun ellipsis (53.8%) constitute the majority of discourse phenomena found in the 119 sentences that require inter-sentential signals for correct translation. In contrast, other categories—tense (4.2%), ambiguity (9.2%) and discourse marker (16.8%)—occur much less frequently.

We next examine how far the useful context tends to be from the cross-lingually ambiguous sentence. Taking $d$ as the sentence distance, the majority of discourse phenomena can be disambiguated based on the nearest context sentence ($d$=1). Specifically, the necessary information for tense, ambiguity, and discourse markers can almost always be found by $d$=1, whereas relevant context for pronoun ellipses and lexical cohesion tends to be more spread out. Hardly any useful information can be found in very distant context ($d$>3).

A significant fraction (40.5%) of sentences in the sampled test set can be translated independently, i.e., without access to inter-sentential information. Correspondingly, we notice that many sentences across document-level data are not lengthy with discourse-level phenomena, but rather simple constructions. Figure 1 indicates that the majority of sentences are relatively short in BWB and IWSLT-17, ranging from 20-50 characters (Chinese) or 10-30 words (French and German).

## 4.2 Context does not help disambiguate certain discourse phenomena.

An implicit assumption in context-aware NMT is that the inclusion of the proper context would influence the model to leverage it to resolve any potential discourse ambiguities. To this end, we investigate different types of discourse phenomena on the

---

[5]Examples for each DM category are in Appendix A.4.

---

| Discourse Phenomena | Freq. | d=1 (%) | d=2 (%) | d=3 (%) | d>3 (%) |
|---|---|---|---|---|---|
| Ellp. Pronoun | 64 | 76.6 | 12.5 | 7.8 | 3.1 |
| Lexical Cohesion | 99 | 56.6 | 23.2 | 13.1 | 7.1 |
| Tense | 5 | 100.0 | 0.0 | 0.0 | 0.0 |
| Ambiguity | 11 | 90.1 | 9.9 | 0.0 | 0.0 |
| Discourse Marker | 20 | 100.0 | 0.0 | 0.0 | 0.0 |
| No Context | 81 | – | – | – | – |

Table 1: Frequency of context-dependent discourse phenomena in a 200-count sample of the BWB test set, and the percentage of cases where relevant context can be found at distance $d = 1, 2, 3, > 3$ sentences.

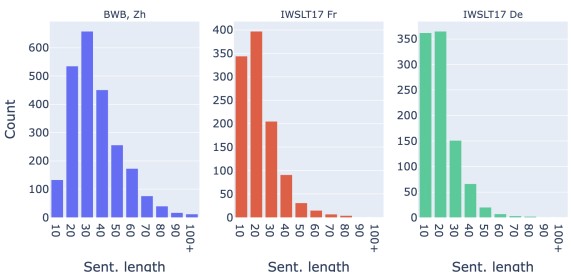

Figure 1: Sentence length distributions on test sets.

BWB test set and show that this premise does not always hold; while pronoun resolution or named entity consistency is often better resolved with the incorporation of context, tense and discourse markers are relatively insensitive to context and yield meager improvement.

### 4.2.1 Pronoun Resolution

We examine two types of pronoun translation: pronoun ellipsis and anaphoric resolution.

**Pronoun ellipsis.** As Chinese is a pro-drop language, pronouns can be freely omitted and are implicitly inferred from surrounding context. In contrast, grammatical and comprehensible translation into English requires that the pronoun be made explicit. To test concatenation-based NMT systems' ability to resolve Chinese-English pronoun ellipsis, we conduct inference on a subset of BWB that contains 519 instances of pronoun ellipsis.

Table 2 indicates that the disambiguation of pronoun ellipsis is particularly responsive to context. Incorporating a single target-side context sentence (the 1-2 setting) improves the BlonDe F1-score from 55.88 to 63.91; adding another source-side context sentence (the 2-2 setting) marginally improves to 65.91. In this scenario, more source-side context may carry useful information, as the 3-1 setting performs the best overall on BlonDe (66.06).

**Anaphoric resolution.** When translating to languages that contain grammatical gender, anaphoric pronouns form another instance of cross-lingual

| | Zh→En | En→De | En→Fr |
|---|---|---|---|
| Setting | BLONDE | CONTRAPRO | BAWDEN |
| 1-1 | 55.88 | 57.75 | 50.00 |
| 1-2 | 63.91 | 55.45 | 64.00 |
| 2-2 | 65.91 | **69.74** | **72.00** |
| 3-1 | **66.06** | – | – |

Table 2: BLONDE evaluation of pronoun translation on the BWB test subset and accuracy for anaphoric pronoun resolution on CONTRAPRO and BAWDEN. The 3-1 setting requires the surrounding context sentences, and therefore cannot be applied to contrastive sets.

| | Consistency (%) | | Acc. (%) |
|---|---|---|---|
| Setting | Person | Non-person | Person |
| 1-1 | 32.34 | 14.67 | **54.55** |
| 1-2 | **49.36** | **21.33** | 51.96 |
| 2-2 | 45.53 | 14.67 | 52.42 |
| 3-1 | 36.17 | 17.33 | 51.15 |

Table 3: Named entity analysis for consistency and accuracy on relevant samples from the BWB test set.

ambiguity. For example, when translating into German, the English pronoun *it* can become either *es*, *sie*, or *er*, depending on the grammatical gender of its referent.

Thus, we also conducted experiments from English to German (En→De) and French (En→Fr), both grammatically gendered languages, and evaluated on the contrastive sets ContraPro (Müller et al., 2018) and Bawden (Bawden et al., 2018), respectively. While Table 2 shows steady improvement for anaphoric resolution on ContraPro, curiously, the 1-2 concatenation-based model exhibits a slight dip compared to its sentence-level counterpart on Bawden. We hypothesize that the small size (200 examples) of the Bawden dataset causes the significant variance in the results.

### 4.2.2 Named Entities

Named entities—real-world objects denoted with proper names—are domain-specific and low-frequency, and thus tend to be absent from bilingual dictionaries (Modrzejewski et al., 2020). Their translations are often either inconsistent (e.g., different target translations for the same source phrase) or inaccurate (with regards to some target reference). In this section, we examine for named entity consistency and accuracy on the annotated BWB test set.

| Type | all | contrast | cause | cond. | conj. | (a)syn. |
|---|---|---|---|---|---|---|
| Count | 2042 | 624 | 361 | 226 | 123 | 705 |
| 1-1 | **55.68** | 58.97 | **40.99** | **71.68** | 47.15 | 56.59 |
| 1-2 | 55.39 | 57.05 | 37.12 | 70.80 | **52.03** | **57.51** |
| 2-2 | 54.99 | 57.05 | 37.12 | 70.80 | 51.21 | 56.79 |
| 3-1 | 53.57 | **59.97** | 37.12 | 65.48 | 43.90 | 54.46 |

Table 4: Accuracy across discourse marker categories and concatenation settings on the BWB test set.

**Consistency.** We extract 780 examples (705 person entities, 75 non-person entities) to construct a *consistency test subset*. Each instance includes a sentence with a named entity that is also mentioned in the preceding sentence. We then measure the frequency at which different context-aware translation models could consistently translate the entity across the two consecutive sentences.

According to Table 3, this task proves to be challenging—no system achieves above-random performance—but the presence of context facilitates consistency as each context-aware setting performs better than the 1-1 baseline on *person* entities (32.34%). Adding target-side context (1-2 and 2-2 settings) appears strictly more helpful. By contrast, source-side context (3-1 setting) results in marginal performance gains relative to the baseline.

**Accuracy.** To explore the frequency at which named entities are *accurately* translated, we next examine the 1734 person entities from the BWB test set. Surprisingly, the sentence-level model is better than context-aware models at correctly translating named entities, with the best accuracy of 54.55% (Table 3). While context is important for ensuring named entity consistency, these findings suggest that adding context may introduce additional noise and do not necessarily lead to more accurate translations. We hypothesize that the dependency on context might hurt the model's downstream performance when the NMT model tries to be consistent with the context translation, which results in a propagation of errors down the sequence.

In addition, when comparing all the results using the *entity* category in BlonDe across the three language pairs in Table 5 and Table 6, it becomes clear that additional context does not meaningfully increase the accuracy of named entity translation.

### 4.2.3 Discourse Marker and Tense

**Discourse makers.** The omission of discourse markers (DM)—particles that signal the type of coherence relation between two segments (Grote and

| | | BLEU | BlonDe | | | | | COMET |
|---|---|---|---|---|---|---|---|---|
| | | | all | pron. | entity | tense | d.m. | |
| XFMR | 1-1 | **20.80**$_{0.20}$ | **38.38**$_{0.38}$ | 72.93$_{1.02}$ | **53.17**$_{1.66}$ | **73.29**$_{0.14}$ | **60.03**$_{0.89}$ | 0.2419$_{0.01}$ |
| | 1-2 | 19.17$_{0.06}$ | 35.77$_{0.28}$ | 74.48$_{1.76}$ | 43.34$_{4.45}$ | 70.70$_{1.15}$ | 57.67$_{1.45}$ | 0.2211$_{0.01}$ |
| | 2-2 | 20.13$_{0.45}$ | 37.63$_{0.40}$ | 76.54$_{0.27}$ | 48.09$_{2.63}$ | 72.93$_{0.35}$ | 59.86$_{0.31}$ | **0.2435**$_{0.01}$ |
| | 3-1 | 19.87$_{0.12}$ | 37.62$_{0.42}$ | **76.59**$_{0.32}$ | 49.76$_{2.64}$ | 72.61$_{0.14}$ | 59.07$_{0.49}$ | 0.2259$_{0.00}$ |
| MEGA | 1-1 | **20.60**$_{0.07}$ | 37.21$_{0.13}$ | 73.08$_{0.26}$ | **49.56**$_{1.29}$ | **73.43**$_{0.27}$ | 60.32$_{0.39}$ | **0.2403**$_{0.00}$ |
| | 1-2 | 20.32$_{0.39}$ | 36.68$_{0.50}$ | 73.56$_{0.16}$ | 46.04$_{1.93}$ | 73.17$_{0.22}$ | **60.35**$_{0.49}$ | 0.2378$_{0.01}$ |
| | 2-2 | 20.34$_{0.27}$ | 36.74$_{0.76}$ | 73.83$_{0.55}$ | 48.78$_{6.80}$ | 73.39$_{0.27}$ | 60.13$_{0.36}$ | 0.2354$_{0.01}$ |
| | 3-1 | 19.87$_{0.25}$ | **37.52**$_{0.38}$ | **76.62**$_{0.49}$ | 49.32$_{1.56}$ | 72.65$_{0.06}$ | 59.23$_{0.23}$ | 0.2299$_{0.01}$ |

Table 5: Automatic metric results on BWB (Zh→En) across different architectures (XFMR and MEGA) and concatenation settings (1-1, 1-2, 2-2, and 3-1). We report average and standard deviations across three runs.

Stede, 1998), e.g., *so*, *because*, *for this reason*—requires context awareness when translating from morphologically poorer languages to morphologically richer ones. Following Jiang et al. (2022), we separate DMs into five categories: *contrast, cause, condition, conjunction,* and *(a-)synchronous*, and examine how different context-aware settings fare with each discourse relation.

As Table 4 shows, the sentence-level (1-1) baseline performs the best across discourse markers in aggregate, and across the *cause* and *condition* categories. The incorporation of context does not significantly improve the accuracy of discourse marker translation; interestingly, the 3-1 setting fares poorly, with the lowest performance across all categories except on *contrast* DMs.

**Tense.** Tense consistency is another extra-sentential phenomenon that requires context for disambiguation, particularly when translating from an analytic source language (e.g., Chinese) to a synthetic target language (e.g., English), wherein tense must be made explicit.[6]

From experimental results on the BWB (Table 5) and IWSLT (Table 6) data,[7] there is minimal variance across all translation settings in the BlonDe scores for tense and DM, suggesting that context is not particularly conducive for any language pair. Tense is generally consistently resolvable, with all models surpassing 70 on Zh→En. As expected, translating from French—a more syn-

thetic language—yields marginally higher BlonDe scores, at over 75. One reason that the BlonDe score for tense may be relatively inflexible across language pairs is that most sentences from the corpora generally adhere to a particular tense, such as past tense in literature, thus diminishing the necessity of context.

### 4.2.4 Is source or target context more helpful?

Fernandes et al. (2021) finds that concatenation-based context-aware NMT models lean on target context more than source context, and that incorporating more context sentences on either side often leads to diminishing returns in performance.

However, according to Table 2-6, this is not universally the case; the effectiveness of target-side versus source-side context is largely dependent on the language pair. Though target-side context often helps with translation consistency, such as preserving grammatical formality across sentences, it does not necessarily guarantee a better translation quality than source-side context (e.g., the 3-1 setting performs best on pronoun translation for French and German according to Table 6, and pronoun ellipsis for Chinese in Table 2).

### 4.3 The context-agnostic baseline performs comparably to context-aware settings.

Experimental results across both the BWB (Table 5) and IWSLT-17 (Table 6) datasets demonstrate that a vanilla 1-1 baseline performs on par with, or even better than its context-aware counterparts on the sentence-level automatic metrics, BLEU and COMET. This suggests that, due to problems with common document-level datasets (e.g., relative lack of contextual signals) (§4.1)

---

[6] In analytic languages, concepts are conveyed through root/stem words with few affixes. Synthetic languages use numerous affixes to combine multiple concepts into single words, incurring a higher morpheme-to-word ratio (O'Grady et al., 1997).

[7] We train on the IWSLT dataset in reverse order (Fr→En and De→En) in order to evaluate with BlonDe.

| | | | BLEU | BlonDe | | | | | COMET |
|---|---|---|---|---|---|---|---|---|---|
| | | | | all | pron. | entity | tense | d.m. | |
| Fr | XFMR | 1-1 | $34.93_{0.15}$ | $52.22_{0.60}$ | $71.64_{0.29}$ | $64.70_{3.55}$ | $75.91_{0.54}$ | $77.78_{0.68}$ | $0.4794_{0.01}$ |
| | | 3-1 | $35.37_{0.15}$ | $52.88_{0.44}$ | $75.42_{1.10}$ | $\mathbf{67.18}_{2.87}$ | $76.09_{0.40}$ | $\mathbf{78.59}_{0.56}$ | $0.4949_{0.02}$ |
| | MEGA | 1-1 | $35.00_{0.72}$ | $51.27_{1.34}$ | $68.16_{2.58}$ | $63.67_{5.71}$ | $75.09_{0.05}$ | $77.53_{0.65}$ | $0.4506_{0.02}$ |
| | | 3-1 | $\mathbf{36.03}_{0.25}$ | $\mathbf{53.37}_{0.19}$ | $\mathbf{77.73}_{3.55}$ | $64.88_{4.53}$ | $\mathbf{76.38}_{0.17}$ | $78.21_{0.71}$ | $\mathbf{0.5095}_{0.02}$ |
| De | XFMR | 1-1 | $30.00_{0.38}$ | $47.76_{0.17}$ | $70.80_{1.75}$ | $65.11_{2.05}$ | $71.58_{0.64}$ | $75.72_{0.38}$ | $0.3250_{0.00}$ |
| | | 3-1 | $30.60_{0.26}$ | $48.23_{0.34}$ | $\mathbf{76.21}_{0.49}$ | $59.44_{1.43}$ | $72.45_{0.44}$ | $75.51_{0.25}$ | $0.3540_{0.01}$ |
| | MEGA | 1-1 | $30.86_{0.25}$ | $48.48_{0.26}$ | $72.52_{3.48}$ | $67.52_{5.36}$ | $\mathbf{73.46}_{2.03}$ | $\mathbf{75.98}_{0.70}$ | $0.3400_{0.01}$ |
| | | 3-1 | $\mathbf{31.21}_{0.37}$ | $\mathbf{49.22}_{0.10}$ | $76.10_{2.88}$ | $\mathbf{68.48}_{4.20}$ | $72.47_{0.27}$ | $75.48_{0.84}$ | $\mathbf{0.3563}_{0.01}$ |

Table 6: Automatic metric results on IWSLT-17 (Fr→En and De→En), on different architectures (XFMR and MEGA) and concatenation settings (1-1 and 3-1). We report average and standard deviations across three runs.

and the inability of sentence-level metrics to capture document-level attributes, context-aware models do not exhibit a meaningful improvement over context-agnostic models at the sentence level.

In terms of document-level improvement, the sentence-level baseline even outperforms context-aware models in select instances, such as when translating named entities (53.17% on Zh, 65.11% on De). There are no notable differences in handling tense and discourse markers across contextual settings, which aligns with our observations in §4.2.3. These results demonstrate that on commonly used datasets, context-aware models also do not significantly improve document-level translation over a sentence-level Transformer baseline.

### 4.4 Advanced model architectures do not meaningfully improve performance.

Motivated by the limitations of the self-attention mechanism on long-range dependency modeling (Tay et al., 2022), recent work has proposed more advanced architectures to better leverage contextual signals into translation (Lupo et al., 2022b; Sun et al., 2022; Wu et al., 2022, 2023). The hypothesis is that as long-range sequence architectures can effectively model longer context windows, they are better-equipped to handle the lengthier nature of document-level translation.

To test this theory, we replace the Transformer (XFMR) attention mechanism with a recently introduced MEGA architecture (Ma et al., 2023), which overcomes several limitations of the Transformer on long-range sequence modeling.[8] As Table 6 shows, MEGA always performs better than XFMR across both the 1-1 and 3-1 settings on the sentence-

[8]We refer to Appendix A.2 for a more detailed discussion on MEGA's design.

level metrics, BLEU and COMET. At the document level, MEGA has the highest overall BlonDe F1-score when translating from both German (53.37 vs. 52.88) and French (49.22 vs. 48.23). While MEGA tends to outscore XFMR on the pronoun and entity categories, there is no significant improvement, if any for tense and discourse marker. Furthermore, MEGA usually starts from a higher sentence-level baseline (except on pronoun resolution for Fr→En); when moving from the sentence-level to the contextual 3-1 setting, MEGA does not show higher relative gains than XFMR.

One potential explanation as to why MEGA performs better on automatic metrics is because it is a stronger model and better at translation overall (Ma et al., 2023), rather than it being able to leverage context in a more useful manner. The lack of improvement in particular discourse categories does not necessarily indicate that existing context-aware models are incapable of handling long-range discourse phenomena. Rather, it suggests that current data may not sufficiently capture the complexities in such situations. As discussed, discourse phenomena are sparse; some of them could not be resolved even with necessary context.

This finding aligns with similar work (Sun et al., 2022; Post and Junczys-Dowmunt, 2023) which also propose that, on existing datasets and under current experimental settings that use sentence-level alignments, the standard Transformer model remains adequate for document-level translation.

### 4.5 There is a need for an appropriate document-level translation metric.

Though BLEU and COMET are both widely used for sentence-level machine translation, they primarily focus on assessing sentence-level transla-

...
We can never go back again, for the past is still too close. The things we have tried to forget would stir again and that sense of fear building up the blind unreasoning panic, now mercifully stilled—might once again become a living companion.
...

— Source paragraph from *Rebecca* (Daphne du Maurier) in English

...
我们再也不能重返故里，这一点确实无疑。过去的影子仍寸步不离地追随着我们。我们竭力想忘记那些往事，把它们抛之脑后，但他们随时都会重新浮现。那种惊恐、内心里惶惶不安的感觉发展到最后，就会变成盲目且不可理喻的慌乱。谢天谢地，眼下我们心境平和，但那种感觉很可能会以某种不可预见的方式重现，又和以前一样跟我们朝夕相伴。
...

— Corresponding human-translated paragraph in Chinese

Figure 2: An example of paragraph-to-paragraph translation. Aligned sentences are underlined in the same color. Highlighted parts are added by translators and do not have a corresponding source segment.

tion quality, and do not adequately encapsulate discourse-level considerations. Contrastive sets are a more discourse-oriented means towards evaluating document-level translation quality, but they too contain shortcomings. First, contrastive sets are not generalizable beyond a particular discourse phenomena and language pair, and the curation of these sets is both time- and labor-intensive. Furthermore, contrastive sets evaluate in a discriminative manner—by asking the model to rank and choose between correct and incorrect translation pairs—which is at odds with, and does not gauge, the MT model's generative capacity. Post and Junczys-Dowmunt (2023) (concurrent work) proposes a generative version of contrastive evaluation, and finds that this paradigm is able to make a finer-grained distinction between document-level NMT systems.

The recently proposed BlonDe (Jiang et al., 2022) score, which calculates the similarity measure of discourse-related spans in different categories, is a first step towards better automatic document-level evaluation. However, BlonDe requires the source language's data to be annotated with discourse-level phenomena, and its applicability is restricted to language pairs in which the target language is English.

Finally, incorporating pre-trained models into metrics is another promising direction. To this end, Vernikos et al. (2022) present a novel approach for extending pre-trained metrics such as COMET to incorporate context for document-level evaluation, and report a better correlation with human preference than BlonDe. Nevertheless, the incorporation of pre-trained models raises the issue of metric interpretability, yielding opaque numbers with no meaningful linguistic explanations. Thus, we note the need to develop more robust, automatic, and interpretable document-level translation metrics.

# 5 PARA2PARA Translation

A recurrent theme throughout our analyses is that existing datasets are not conducive to meaningful context usage in document-level translation. The majority of datasets used in the literature of document-level NMT are aligned at the sentence level, which is artificial in design and not reflective of how documents are translated in practice.

As such, paragraph-level parallel data (Figure 2) may be more suited for document-level NMT and provide richer contextual training signals. Recent work have turned toward literary translation as a challenging, realistic setting for document-level translation (Zhang and Liu, 2020; Thai et al., 2022; Karpinska and Iyyer, 2023), given that literary texts typically contain complex discourse structures that mandate a document-level frame of reference. As Figure 2 illustrates, sentence alignment is not always feasible when translating literature. Karpinska and Iyyer (2023) finds that language models can effectively exploit document-level context and cause fewer discourse-level translation errors based on human evaluation, when the *paragraph* is taken as the minimal discourse-level unit.

To promote future research on document-level translation in a realistic setting, we collect professional English and Chinese translations of classic novels, and format the data by manually correcting paragraph-level alignments. The PARA2PARA dataset consists of 10,545 parallel paragraphs across six novels from the public domain.[9] To our knowledge, the only other paragraph-aligned, parallel dataset sourcing from the literary domain is PAR3 (Thai et al., 2022), which uses Google Translate and fine-tuned GPT-3 (Brown et al., 2020) to automatically generate reference translations. In

---

[9]Data and preprocessing details are in Appendix B.1.

| Domain | Pre-training | BLEU | BlonDe | | | | | COMET |
|---|---|---|---|---|---|---|---|---|
| | | | all | pron. | entity | tense | d.m. | |
| Closed | NONE | $1.37_{0.06}$ | $8.44_{0.39}$ | $47.28_{2.18}$ | $18.73_{6.49}$ | $41.22_{1.48}$ | $16.87_{3.60}$ | $0.3949_{0.00}$ |
| | XFMR$_{Big}$ | $16.00_{0.10}$ | $35.36_{0.19}$ | $79.16_{0.4}$ | $52.33_{0.10}$ | $72.47_{0.46}$ | $60.63_{0.42}$ | $0.7339_{0.00}$ |
| | LIGHTCONV$_{Big}$ | $\mathbf{16.87}_{0.06}$ | $\mathbf{36.70}_{0.14}$ | $\mathbf{79.28}_{0.28}$ | $\mathbf{55.38}_{1.27}$ | $\mathbf{72.80}_{0.09}$ | $\mathbf{61.66}_{0.61}$ | $\mathbf{0.7409}_{0.00}$ |
| | MBART25 | $15.63_{0.25}$ | $35.37_{0.20}$ | $78.72_{0.27}$ | $54.04_{0.49}$ | $72.01_{0.12}$ | $60.59_{0.62}$ | $0.7385_{0.00}$ |
| Open | NONE | $0.73_{0.32}$ | $1.82_{0.21}$ | $48.12_{5.54}$ | $0.00_{0.00}$ | $39.27_{2.11}$ | $13.91_{3.90}$ | $0.3587_{0.02}$ |
| | XFMR$_{Big}$ | $\mathbf{9.17}_{0.67}$ | $25.35_{1.05}$ | $72.20_{0.48}$ | $32.54_{2.18}$ | $67.17_{0.59}$ | $\mathbf{51.83}_{1.00}$ | $0.7003_{0.01}$ |
| | LIGHTCONV$_{Big}$ | $8.60_{0.10}$ | $\mathbf{25.48}_{0.10}$ | $\mathbf{72.50}_{0.43}$ | $\mathbf{38.83}_{0.57}$ | $\mathbf{67.40}_{0.35}$ | $51.79_{0.92}$ | $\mathbf{0.7027}_{0.00}$ |
| | MBART25 | $7.97_{0.06}$ | $22.41_{0.71}$ | $72.24_{0.61}$ | $20.07_{4.17}$ | $67.25_{0.40}$ | $50.52_{0.84}$ | $0.7012_{0.00}$ |

Table 7: Baseline translation results on the Zh→En PARA2PARA dataset. **Bold** denotes best performance.

contrast, the source and target paragraphs in our dataset are culled from professional translations.[10]

We then benchmark the dataset under two experimental settings for Zh→En translation: i). a standard **closed-domain** setup, in which both the training and testing data are sourced from the same novels; ii). a more challenging **open-domain** setup, wherein two novels are held and used as only the test set. We experiment with training a Transformer-based model on PARA2PARA data from scratch (NONE), as well as incorporating pre-trained baselines, in which the model is first trained on the sentence-level WMT17 Zh-En dataset (Bojar et al., 2017), before further fine-tuning on the PARA2PARA data, using the following backbone architectures:

- XFMR$_{Big}$ (Vaswani et al., 2017), the Transformer-BIG.
- LIGHTCONV$_{Big}$ (Wu et al., 2019), which replaces the self-attention modules in the Transformer-BIG with fixed convolutions.
- MBART25 (Liu et al., 2020), which is pre-trained on 25 languages at the document level.

Table 7 shows preliminary baseline results on BLEU, BlonDe, and COMET.[11] In the NONE setting, the Transformer's relatively low performance and incoherent output underscores the difficulty of training from scratch on the PARA2PARA corpus, due to two reasons—the inherent difficulty of training on paragraph-level, longer-sequence data, and the limited dataset size (especially relative to that of sentence-level MT datasets). To disentangle the two factors, we report additional baselines that leverage pre-training to offset the

issue of low-domain data; all of them exhibit a marked performance improvement over the NONE setting, attesting to the challenging constitution of paragraph-to-paragraph translation.

On the closed-domain setting, LIGHTCONV$_{Big}$ yields the highest score across all three metrics. Open-domain results are mixed: as expected, scores are lower across the board as this setting is challenging. XFMR$_{Big}$ has the best BLEU and discourse marker F1-score on BlonDe, although all pre-training baselines perform similarly. LIGHTCONV$_{Big}$ performs the best on pronoun, entity, and tense on BlonDe and has the highest COMET score.

## 6 Conclusion

Despite machine-human parity at the sentence level, NMT still lags behind human translation on long collections of text, motivating the need for context-aware systems that leverage signals beyond the current sentence boundary. In this work, we highlight and discuss key obstacles that hinder momentum in context-aware NMT. We find that training signals that improve document-level discourse phenomena occur infrequently in surrounding context, and that most sentences can be accurately translated in isolation. Another challenge is that context benefits the resolution of some discourse phenomena over others. A context-agnostic Transformer baseline is already competitive against context-aware settings, and replacing the Transformer's self-attention mechanism with a more complex long-range mechanism does not significantly improve translation performance. We also note the need for a generalizable document-level translation metric. Finally, we make the case for paragraph-aligned translation, and release a new PARA2PARA dataset, alongside baseline results, to encourage further efforts in this direction.

---

[10]Another distinction is that the Zh-En split in PAR3 sources from ancient novels in Classical Chinese (which is different from the modern language) and consists of 1320 paragraphs.

[11]Example translations are in Appendix B.2.

## 7 Limitations

Several limitations restrict the scope of this work. To begin, our choice of languages in this study—English, Chinese, French, German—is non-exhaustive, and it is possible that our findings would not generalize well to scenarios that involve low-resourced languages or distant language pairs. In particular, a significant portion of our investigation on discourse relations that necessitate context for proper disambiguation targets the Chinese-English BWB test set, which is the only public dataset that has been manually annotated with this type of information. Some of the discourse phenomena that we consider may not occur as frequently in other languages. While this work is a preliminary step that sheds light on the current nature of *data* that drives context-aware neural machine translation, future directions could entail extending similar analysis to other languages or discourse phenomena (e.g., the disambiguation of deixis when translating from Russian to English (Voita et al., 2019)).

Another restriction is that this work only examines concatenation-based architectures, which tend to be conceptually simple, effective, and hence subject to widespread adoption in recent years (Fernandes et al., 2021). While the purported advantages of multi-encoder NMT models are mixed (Li et al., 2020), for comprehensiveness, it would be insightful to examine whether they behave differently relative to concatenation-based systems under our experimental setup. Other potential avenues for exploration entail loss-based approaches to context-aware neural machine translation, such as context discounting (Lupo et al., 2022b) or contrastive learning-based schemas (Hwang et al., 2021).

Lastly, although the PARA2PARA dataset may pose as a more natural setting for context-aware translation, it is considerably smaller than other document-level datasets. Given that the small scale of training data is a prevalent issue in context-aware neural machine translation (Sun et al., 2022), future efforts could focus on expanding this dataset (as it is easier to source paragraph-aligned parallel translations in the wild than sentence-level ones) or moving beyond the literary domain.

## Acknowledgements

We thank the anonymous reviewers for their constructive feedback in improving this work. JH is supported by an NSF Graduate Research Fellowship. This research is supported in part by the Office of the Director of National Intelligence (ODNI), Intelligence Advanced Research Projects Activity (IARPA), via the HIATUS Program contract #2022-22072200006. The views and conclusions contained herein are those of the authors and should not be interpreted as necessarily representing the official policies, either expressed or implied, of ODNI, IARPA, or the U.S. Government. The U.S. Government is authorized to reproduce and distribute reprints for governmental purposes notwithstanding any copyright annotation therein.

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

# Appendix

## A  Implementation Details

### A.1  Training

We train all models on the `fairseq` framework (Ott et al., 2019). Following Vaswani et al. (2017); Fernandes et al. (2021), we use the Adam optimizer with $\beta_1 = 0.9$ and $\beta_2 = 0.98$, dropout set to 0.3, an inverse square root learning rate scheduler with an initial value of $10^{-4}$, and the warm-up step set to 4000. We run inference on the validation set and save the checkpoint with the best BLEU score. We compute all BLEU scores using the `sacreBLEU` toolkit (Post, 2018).[12] Wherever possible, we report the average and standard deviation across three randomly seeded runs.

### A.2  Models

**Transformer**   The Transformer (Vaswani et al., 2017) is an encoder-decoder architecture that relies on a self-attention mechanism, in which every position of a single sequence relates to one another in order to compute a representation of that sequence. An $n$-length output sequence of $d$-dimensional representations $\boldsymbol{Y} \in \mathcal{R}^{n \times d}$ can be computed from an input sequence of $d$-dimensional representations $\boldsymbol{X} \in \mathcal{R}^{n \times d}$ as follows:

$$\boldsymbol{Y} = \text{Attn}(\boldsymbol{X}) = f\left(\frac{\boldsymbol{Q}\boldsymbol{K}^T}{\tau(\boldsymbol{X})}\right)\boldsymbol{V} \qquad (3)$$

$\boldsymbol{Q}$, $\boldsymbol{K}$, and $\boldsymbol{V}$ are sequences of queries, keys, and values, respectively, with learnable weights and biases. Here, $f(\cdot)$ is an attention function, most commonly set to softmax, and $\tau$ is a correspondent scaling term. We use the Transformer base version across all experiments, which consists of 6 encoder layers, 6 decoder layers, a model dimension of 512, and an FFN hidden dimension of 2048.

**MEGA**   The recently introduced MEGA (Moving Average Equipped Gated Attention) (Ma et al., 2023) architecture solves for two limitations of the traditional Transformer, which have long since resulted in sub-optimal performance on long-sequence tasks: a weak inductive bias, and a quadratic computational complexity. This mechanism applies a multi-dimensional, damped exponential moving average (Hunter, 1986) (EMA) to

---

[12]The sacreBLEU signature is BLEU+case.mixed+lang.src-tgt+numrefs.1+smooth.exp+{test-set}+tok.13a.

a single-head gated attention, in order to preserve inductive biases. MEGA serves as a drop-in replacement for the Transformer attention mechanism, and full details can be found in (Ma et al., 2023). MEGA is of comparable size to the Transformer, with 6 encoder and 6 decoder layers, a model dimension of 512, and an FFN hidden dimension of 1024, alongside an additional shared representation dimension (128), value sequence dimension (1024), and EMA dimension (16).

In total, the Transformer architecture is around 65M parameters; the MEGA architecture is around 67M parameters.

### A.3 Data

For the En↔Fr and En↔De language pairs, we train on the IWSLT17 (Cettolo et al., 2012) datasets, which contain document-level transcriptions and translations culled from TED talks. The test sets from 2011-2014 are used for validation, and the 2015 test set is held for inference. For Zh→En, we use the BWB (Jiang et al., 2023) dataset, which consists of Chinese webnovels.

Data for each language pair is encoded and vectorized with byte-pair encoding (Sennrich et al., 2016) using the SentencePiece (Kudo and Richardson, 2018) framework. We use a 32K joint vocabulary size for Zh→En, and a 20K vocabulary size for the other language pairs.

Full corpus statistics are in Table 8.

| Dataset | Lg. Pair | Train | Valid | Test |
|---|---|---|---|---|
| BWB | Zh→En | 9576566 | 2632 | 2618 |
| WMT17 | Zh→En | 25134743 | 2002 | 2001 |
| IWSLT17 | En↔Fr | 232825 | 5819 | 1210 |
| IWSLT17 | En↔De | 206112 | 5431 | 1080 |

Table 8: Sentence counts across parallel datasets.

### A.4 Evaluation

**Annotated BWB test set.** Some manually annotated paragraphs from the BWB test set can be found in Table 9, which is used in the discourse phenomena analysis.

**Discourse marker categories.** Following (Jiang et al., 2022), we categorize discourse markers into the following:

- **Contrast**: but, while, however, although, though, yet, whereas, in contrast, by comparison, conversely

- **Cause**: so, thus, hence, as a result, therefore, thereby, accordingly, consequently, for this reason

- **Condition**: if, as long as, provided that, assuming that, given that

- **Conjunction**: also, in addition, moreover, additionally, besides, else, plus, furthermore

- **(A)synchronous**: when, after, then, before, until, after, once, after, next

**Contrastive set examples.** Contrastive evaluation examples for anaphoric pronoun resolution are in Table 10. Following standard practice, the model is evaluated in a *discriminative* manner: rather than generating translated sequences, the model is provided with the previous sentence as context, and is asked to choose the current sentence with the correct pronoun from the incorrect ones.

## B PARA2PARA Translation

### B.1 Data and Preprocessing

We gather the Chinese and English versions of six novels within the public domain, which are freely available online (Table 11). Prior to the tokenization step, we normalize punctuation and segment Chinese sentences using the open-sourced Jieba package. English sentences are tokenized using the Moses toolkit (Koehn et al., 2007). We employ byte-pair encoding (Sennrich et al., 2016) for subword tokenization.

In the open-domain setting, *A Tale of Two Cities* and *Twenty Thousand Leagues Under the Seas* are withheld as the test set.

### B.2 Translation Examples

Translation examples on the PARA2PARA dataset are in Figure 3.

### B.3 LLM Evaluations

Large Language Models (LLMs) (e.g., Chat-GPT (OpenAI, 2022)) have recently accrued a great deal of mainstream and scientific interest, as they are found to maintain considerable fluency, consistency, and coherency across multiple NLP tasks, including document-level NMT (Wang et al., 2023)(concurrent work). To investigate how LLMs would fare on the PARA2PARA dataset, we also obtain translations using GPT-3.5 (gpt-3.5-turbo), a commercial, black-box LLM. Table 12 shows GPT-3.5's performance alongside that of the three

| | | |
|---|---|---|
| <PER, T,1>{Qiao Lian} clenched <O,1>{her} fists and lowered <O,1>{her}head. | | |

<PER, T,1>{Qiao Lian} clenched <O,1>{her} fists and lowered <O,1>{her}head.
Actually, <P,2>{he} was right.
<O,1>{She} was indeed an idiot, as only an idiot would believe that they could find true love online.
<P,1>{She} curled <P,1>her} lips and took a deep breath. ... <ORG, T, 3>{WeChat} account.
<Q,1><PER, T,1>{Qiao Lian}: "What happened?" <\Q>

<PER,T,18>{Song Cheng} was extremely nervous and followed <P,10>{him}.
<PER,T,10>{Shen Liangchuan} walked forward, one step at a time,
until <O,10>{he} reached the front of <FAC,N,19>{the room}.
<PER,T,12>{Wang Wenhao} was currently ingratiating <O,12>{himself} with <PER,N,20>{a C-list celebrity}.
<PER,N,20>{The celebrity} asked, <Q,20>"Hey, I heard that you beat <PER,N,21>{a paparazzi}?"<\Q>
<Q,12>"Yeah, <PER,N,21>{the paparazzi} nowadays are so disgusting.
I have wanted to teach <P,21>{them} a lesson myself for some time now!"<\Q>
<Q,20>"Are not you afraid of becoming an enemy of <P,21>{them}?"<\Q>

Table 9: Annotated paragraphs from the BWB test set.

| | | Context Sentence | Current Sentence |
|---|---|---|---|
| **ContraPro (En-De)** | src | There were spring nights. | Through open windows it came in, dancing. |
| | tgt | Es gab FrÜhlingsnächte. | 1. Bei offenen Fenstern tanzt es herein. |
| | | | 2. Bei offenen Fenstern tanzt sie herein. |
| | | | 3. Bei offenen Fenstern tanzt er herein. |
| **Bawden (En-Fr)** | src | The next Saturday night. | Only one road led to the Huseby summer farm, and it passed right by the main farm. |
| | tgt | Le dimanche soir suivant. | 1. Une seule route conduisait à la ferme d'été des Huseby, et elle passait devant la grande ferme. |
| | | | 2. Une seule route conduisait à la ferme d'été des Huseby, et il passait devant la grande ferme. |

Table 10: Examples from the ContraPro and Bawden contrastive evaluation sets. Highlighted pronouns in the current sentence require the preceding context sentence for proper disambiguation.

pre-trained baselines, for reference. This experiment is similar to that of Karpinska and Iyyer (2023), who test GPT-3.5 on paragraphs from recently-published literary translations, and show that while LLMs can provide better paragraph-level translation (as they are better-equipped to handle long context), there are nevertheless critical translation errors that a human translator would be able to avoid. Given that OpenAI did not disclose the composition of ChatGPT's training data, it is likely that there may be data leakage from pre-training (especially as our dataset is sourced from public-domain data). Thus, we do not believe these results represent a fair comparison with the pre-training baselines; we report them for the sake of comprehensiveness.

## B.4 Pre-trained baseline performance

To investigate how fine-tuning on the PARA2PARA dataset affects the baselines' performance, we evaluate the pre-trained baselines on the same test set without any training on the PARA2PARA corpus. As Table 13 illustrates, all three baselines exhibit significantly worse performance across the board (✗), and improve after fine-tuning (✓).

| Title | Author | Year | # Paras. | APL |
|---|---|---|---|---|
| Gone with the Wind | Margaret Mitchell | 1936 | 3556 | 143 |
| Rebecca | Daphne du Maurier | 1938 | 1237 | 157 |
| Alice's Adventure in Wonderland | Lewis Carroll | 1865 | 218 | 144 |
| Foundation | Isaac Asimov | 1951 | 3413 | 76 |
| A Tale of Two Cities | Charles Dickens | 1859 | 696 | 225 |
| Twenty Thousand Leagues Under the Seas | Jules Verne | 1870 | 1425 | 117 |

Table 11: Corpus information for the PARA2PARA dataset. APL = average paragraph length in tokens.

| Domain | Pre-training | BLEU | BlonDe | | | | | COMET |
|---|---|---|---|---|---|---|---|---|
| | | | all | pron. | entity | tense | d.m. | |
| Closed | GPT-3.5 | 11.60 | 28.29 | **83.21** | 35.63 | **73.98** | 63.82 | **0.7644** |
| | XFMR$_{Big}$ | 16.00 | 35.36 | 79.16 | 52.33 | 72.47 | 60.63 | 0.7339 |
| | LIGHTCONV$_{Big}$ | **16.87** | **36.70** | 79.28 | **55.38** | 72.80 | 61.66 | 0.7409 |
| | mBART25 | 15.63 | 35.37 | 78.72 | 54.04 | 72.01 | 60.59 | 0.7385 |
| Open | GPT-3.5 | **11.90** | **27.86** | **86.02** | 30.64 | **75.71** | **66.93** | **0.7648** |
| | XFMR$_{Big}$ | 9.17 | 25.35 | 72.20 | 32.54 | 67.17 | 51.83 | 0.7003 |
| | LIGHTCONV$_{Big}$ | 8.60 | 25.48 | 72.50 | **38.83** | 67.40 | 51.79 | 0.7027 |
| | mBART25 | 7.97 | 22.41 | 72.24 | 20.07 | 67.25 | 50.52 | 0.7012 |

Table 12: GPT-3.5 evaluations on the PARA2PARA dataset.

| Domain | Pre-training | Fine-tuning | BLEU | BlonDe | | | | | COMET |
|---|---|---|---|---|---|---|---|---|---|
| | | | | all | pron. | entity | tense | d.m. | |
| Closed | XFMR$_{Big}$ | ✗ | **7.30** | **25.42** | **67.41** | **37.24** | 60.82 | **54.06** | 0.6662 |
| | LIGHTCONV$_{Big}$ | ✗ | 6.30 | 23.30 | 58.19 | 34.32 | 56.16 | 47.57 | 0.6487 |
| | mBART25 | ✗ | 6.70 | 21.35 | 63.56 | 22.10 | **60.96** | 47.52 | **0.6699** |
| Open | XFMR$_{Big}$ | ✗ | **4.70** | **21.22** | **59.81** | 35.45 | 54.31 | **44.12** | 0.6342 |
| | LIGHTCONV$_{Big}$ | ✗ | 4.20 | 20.78 | 53.35 | **38.94** | 49.80 | 39.96 | 0.6160 |
| | mBART25 | ✗ | 4.10 | 18.61 | 55.89 | 26.27 | **54.39** | 39.35 | **0.6407** |
| Closed | NONE | ✓ | 1.37$_{0.06}$ | 8.44$_{0.39}$ | 47.28$_{2.18}$ | 18.73$_{6.49}$ | 41.22$_{1.48}$ | 16.87$_{3.60}$ | 0.3949$_{0.00}$ |
| | XFMR$_{Big}$ | ✓ | 16.00$_{0.10}$ | 35.36$_{0.19}$ | 79.16$_{0.4}$ | 52.33$_{0.10}$ | 72.47$_{0.46}$ | 60.63$_{0.42}$ | 0.7339$_{0.00}$ |
| | LIGHTCONV$_{Big}$ | ✓ | **16.87$_{0.06}$** | **36.70$_{0.14}$** | **79.28$_{0.28}$** | **55.38$_{1.27}$** | **72.80$_{0.09}$** | **61.66$_{0.61}$** | **0.7409$_{0.00}$** |
| | mBART25 | ✓ | 15.63$_{0.25}$ | 35.37$_{0.20}$ | 78.72$_{0.27}$ | 54.04$_{0.49}$ | 72.01$_{0.12}$ | 60.59$_{0.62}$ | 0.7385$_{0.00}$ |
| Open | NONE | ✓ | 0.73$_{0.32}$ | 1.82$_{0.21}$ | 48.12$_{5.54}$ | 0.00$_{0.00}$ | 39.27$_{2.11}$ | 13.91$_{3.90}$ | 0.3587$_{0.02}$ |
| | XFMR$_{Big}$ | ✓ | **9.17$_{0.67}$** | 25.35$_{1.05}$ | 72.20$_{0.48}$ | 32.54$_{2.18}$ | 67.17$_{0.59}$ | **51.83$_{1.00}$** | 0.7003$_{0.01}$ |
| | LIGHTCONV$_{Big}$ | ✓ | 8.60$_{0.10}$ | **25.48$_{0.10}$** | **72.50$_{0.43}$** | **38.83$_{0.57}$** | **67.40$_{0.35}$** | 51.79$_{0.92}$ | **0.7027$_{0.00}$** |
| | mBART25 | ✓ | 7.97$_{0.06}$ | 22.41$_{0.71}$ | 72.24$_{0.61}$ | 20.07$_{4.17}$ | 67.25$_{0.40}$ | 50.52$_{0.84}$ | 0.7012$_{0.00}$ |

Table 13: Ablation study on the effect of fine-tuning on the Zh→En PARA2PARA dataset. ✗ no fine-tuning; ✓ denotes fine-tuning. **Bold** denotes best performance.

| | |
|---|---|
| **Source** | 我的脸色清楚地透露了我这种想法,但尼摩船长不说:什么,只请我跟着他走俄就像不顾一切地听天由命的人一样跟着他.我们到了饭厅,早餐早就摆好在那里了."阿龙纳斯先生",船长对我说",我请您用饭,不要客:气.我们一边吃饭,一边谈话.尽管我答应您可以去林中散步,但我并没有向您保证可以在林中碰到一家饭馆.所以请您尽量吃,就像一个要很迟才能回来吃午饭的人一 |
| **Reference** | These thoughts were clearly readable on my face ; but Captain Nemo remained content with inviting me to follow him , and I did so like a man resigned to the worst . We arrived at the dining room , where we found breakfast served . " Professor Aronnax , " the captain told me , " I beg you to share my breakfast without formality . We can chat while we eat |
| **Open Domain** Base | I tried to think of an excuse. I knew he did not want to lunch with me. It was his form of courtesy. I should ruin his meal. I |
| **Open Domain** Base+Transformer-WMT17 | My face told me so clearly, but Captain Nemo did not say: "What, just ask me to go with him and follow him like a man who has no choice but to listen to his fate. We are in the dining room, and breakfast has been arranged there." Mr. Aronnus, "said the captain," I ask you to take your dinner, and don't be invidious. We talk as we eat. I don't give you any |
| **Open Domain** Base+LightConv-WMT17 | My face told me this clearly, but Captain Nemo did not say, 'What, to ask me to follow him is like following him like a man who listens desperately to fate. We are in the dining room, and breakfast is there.' Mr Aronas, 'said the captain to me,' I ask you to have dinner, not to be polite. We have dinner, and we talk. Though I promised you I might go out for a walk in |
| **Open Domain** Base+mBART-WMT17 | My face told me so clearly, but Captain Nemo didn't say anything. He asked me to follow him, just like a man who listens desperately to his own will. We went to the dining-room, and breakfast was already ready. "Professor Aronnax," the captain told me, "I ask you to eat well. Don't be rude. We dine and talk. Although I promise you to go for a walk in the |
| **Closed Domain** Base | Captain Nemo gave me a few words. "Well, sir, sir, sir, sir, sir, sir, sir, sir, sir, sir, sir, sir, sir, sir." "I replied," I replied, "I replied, sir." |
| **Closed Domain** Base+Transformer-WMT17 | My face betrayed my idea clearly, but Captain Nemo said nothing: Let me follow him, and follow him like a desperate man. We were in the dining-room, and breakfast had been arranged there. "Mr. Aronnax," the captain said to me, "I beg |
| **Closed Domain** Base+LightConv-WMT17 | My face told me so clearly, but Captain Nemo did not say anything. Just ask me to follow him, just like a man who obeys orders recklessly. We arrived at the dining-room, and breakfast was ready. "Professor Aronnax," the captain told me, "I'll treat you to lunch, not to be offended. We'll have dinner together and talk. Though I promised you I could go for a stroll in |
| **Closed Domain** Base+mBART-WMT17 | My face told me so plainly, but Captain Nemo didn't say anything. He merely asked me to follow him, as if he were a man who listened desperately to his will. We arrived in the dining-room, and breakfast was ready for us. "Professor Aronnax," the captain told me, "I ask you to eat well. Don't be rude. We dine and talk. Though I promise you you a walk in the |

Figure 3: An example of PARA2PARA translation across open-domain and closed-domain settings.