# OpenReview forum: "Challenges in Context-Aware Neural Machine Translation"
_EMNLP/2023/Conference — EMNLP 2023 Main_

### Official Review · Reviewer_dPV2 · 2023-07-19

**Soundness:** 3

**Excitement:**

3: Ambivalent: It has merits (e.g., it reports state-of-the-art results, the idea is nice), but there are key weaknesses (e.g., it describes incremental work), and it can significantly benefit from another round of revision. However, I won't object to accepting it if my co-reviewers champion it.

**Paper Topic And Main Contributions:**

This paper takes an in-depth look into the testset of document-level MT, it finds that 1) existing document-level corpora contain a sparse number of discourse phenomena that rely upon inter-sentential context to be accurately translated, and 2) context is less helpful for tense and discourse markers. The paper suggests to perform paragraph-to-paragraph translation and construct a new paragraph-aligned Chinese-English dataset.

**Questions For The Authors:**

A, where does the target context come from during inference? the model decoding results or the gold reference? How do you close the gap between training and inference if you use the decoded context?

thanks for the response, with Question A, I want to know whether the target-side input contexts are noised or not during training.

**Reasons To Accept:**

The analysis on the context-aware translation testset points out key obstacles for document-level MT studies (existing document-level corpora contain a sparse number of discourse phenomena that rely upon inter-sentential context to be accurately translated and context is less helpful for tense and discourse markers). Based on these findings, the paper suggests to perform paragraph-to-paragraph translation and provides a paragraph-aligned Chinese-English dataset.

**Reasons To Reject:**

A, some of the findings are already proposed in previous studies.

B, paragraph-to-paragraph translation is simple in practice and lacks novelty.

**Reproducibility:**

4: Could mostly reproduce the results, but there may be some variation because of sample variance or minor variations in their interpretation of the protocol or method.

**Reviewer Confidence:**

4: Quite sure. I tried to check the important points carefully. It's unlikely, though conceivable, that I missed something that should affect my ratings.

---

> ### Author Rebuttal · Authors · 2023-08-28
>
> We thank R3 for their time and feedback! We are encouraged that R3 finds our analysis to point out “key obstacles for document-level MT studies,” and respond point-by-point to their feedback below: \
> \
> **“Some of the findings are already proposed in previous studies.”**
>
> Some findings have indeed surfaced in previous studies, but the purpose of this work is to explore such claims in greater and more rigorous detail across a standardized experimental set-up. Our experiments either corroborate or refute such commonly-accepted beliefs, yielding new insights that we believe to be useful to the scientific community.
>
> Given that R3 did not provide specific references to back up this claim, we refer to our response to R1, in which we provide some concrete examples wherein our paper adds quantitative analysis to common perceptions in this field. For example,
>
> - *Data sparsity*: this is first posited by Lupo et al., 2022 [1], though they do not perform any data investigation and only focus on multi-encoder architectures. Our analysis is the first to quantitatively show the presence of context sparsity across document-level data (BWB, IWSLT).
>
> - *There is a need for an appropriate document-level translation metric*: Our study is the first to show that existing metrics are only selectively helpful based on discourse category (Sect. 4.2). We additionally note several limitations in Sect. 4.5.
>
> To date, no other work has attempted a comprehensive analysis of the obstacles that impede progress in context-aware NMT.
>
> [1] [Lupo et al. Divide and rule: Effective pre-training for context-aware multi-encoder translation models. Annual Meeting of the Association for Computational Linguistics. 2022](https://arxiv.org/abs/2103.17151) \
> \
> \
> **“Paragraph-to-paragraph translation is simple in practice and lacks novelty”**
>
> Though paragraph-to-paragraph translation may be conceptually simple, translating literary works presents a significantly greater challenge compared to standard MT tasks [1]. Moreover, paragraphs from literary texts provide richer contextual signals. We empirically show in Table 7 that this task is in fact quite challenging, as even powerful pre-trained models struggle on this task, and the performance discrepancy becomes even more apparent under open-domain settings.
>
> Furthermore, to the best of our knowledge, our proposed dataset (Para2Para) is the first paragraph-aligned, parallel translation dataset in which source and target paragraphs are harvested from high-quality professional literary translations. The only other paragraph-aligned, literary translation dataset that we are aware of is Par3 [1],  which uses fine-tuned GPT-3 to produce target translations, and whose Zh-En split sources from ancient Chinese novels written in Classical Chinese (which is very different in from the modern language).
>
> [1] [Thai et al. Exploring Document-Level Literary Machine Translation with Parallel Paragraphs from World Literature. Conference on Empirical Methods in Natural Language Processing. 2022.](https://arxiv.org/pdf/2210.14250) \
> \
> **3A.**\
> Following previous work on context-aware NMT [1, 2, 3], we use the preceding decoded sentence  as target context during inference for a fair and realistic comparison (as one would not have access to gold target data when translating in real-world scenarios) .
> There is certainly a performance difference between leveraging either the decoded translation versus the gold reference during test-time translation; however, closing the gap is beyond the scope of this work. \
> \
> [1] [Fernandes et al. Measuring and Increasing Context Usage in Context-Aware Machine Translation. ArXiv abs/2105.03482. 2021: n. pag.](https://arxiv.org/abs/2105.03482) \
> [2] [Yin et al. Do Context-Aware Translation Models Pay the Right Attention? Annual Meeting of the Association for Computational Linguistics. 2021](https://arxiv.org/abs/2105.06977) \
> [3] [Lupo et al. Divide and rule: Effective pre-training for context-aware multi-encoder translation models. Annual Meeting of the Association for Computational Linguistics. 2022](https://arxiv.org/abs/2103.17151)

---

### Official Review · Reviewer_LPga · 2023-08-03

**Soundness:** 4

**Excitement:**

3: Ambivalent: It has merits (e.g., it reports state-of-the-art results, the idea is nice), but there are key weaknesses (e.g., it describes incremental work), and it can significantly benefit from another round of revision. However, I won't object to accepting it if my co-reviewers champion it.

**Paper Topic And Main Contributions:**

This paper talks about context-aware (or document-level) neural machine translation.

On a high level, there are two major parts in the paper.
The first part analyze the efficacy and chanllenges of context-aware NMT, providing detailed information on aspects like how often document-level context is useful, how well can different types of discourse phenomena be resolved, and how existing document-level systems compare to a strong sentence-level baseline.
The second part introduced a new dataset, which is a paragraph-aligned translation dataset that are based on open-domain books. The dataset is quite challenging, because of the relatively small size, the translation style, and the occasionally added translations without corresponding source sentences. The authors also provided baseline results on this dataset.

Overall, the paper is solid, aligned with the interests of EMNLP, and provides value for future research in context-aware NMT.

**Questions For The Authors:**

L603, why do you say it shows difficulty to train on scratch on longer-sequence data? To me, the relatively low performance comes from limited training data. On top of this point, we know LLMs can handle long contexts quite well, maybe you could prompt some LLMs to generate hypotheses on your test sets as well? Would be interesting to see how out-of-the-box "ChatGPT" compares with pre-trained and finetuned in-domain document-level NMT systems.

**Reasons To Accept:**

The paper is well-written. The text is smooth and easy to follow.

The analyses are extensive. Although context-aware translation itself is a hard topic, the authors did a good job disecting the problem, and provided multiple angles in understanding the challenges and the current landscape.

The introduced paragraph-aligned dataset is a nice and challenging resource to boost further research. While comparing scores of automatic metrics is easily envisioned, I can also see people getting interested in proposing better evaluation metrics for such paragragh-aligned test sets.

**Reasons To Reject:**

Solid work, no apparent reasons to reject.
(Although one may argue that the language pairs/model variations are not that extensive, I don't think this point alone justifies a rejection.)

**Reproducibility:**

4: Could mostly reproduce the results, but there may be some variation because of sample variance or minor variations in their interpretation of the protocol or method.

**Reviewer Confidence:**

3: Pretty sure, but there's a chance I missed something. Although I have a good feel for this area in general, I did not carefully check the paper's details, e.g., the math, experimental design, or novelty.

**Typos Grammar Style And Presentation Improvements:**

In Table 7 and the corresponding text describing it, you conclude that it is necesary to pretrain the model the get decent performance on the proposed paragraph-level dataset. For this purpose, I think it also makes sense to include scores for the pre-trained but not finetuned models. Including these results can also eliminate the reasoning that the pre-trained models are already good enough without finetuning.

---

> ### Author Rebuttal · Authors · 2023-08-28
>
> We thank R2 for their time and feedback! We are encouraged that R2 finds the that “the paper is well-written” and “provides value for future research in context-aware NMT,”  that “the analyses are extensive,” and that our Para2Para dataset is a “nice and challenging resource to boost further research.”
>
> We respond point-by-point to R2’s feedback below: \
> \
> **“Language pairs/model variations are not that extensive.”**
>
> We agree that our analysis could definitely benefit from more language pairs / model variations. Given that many popular document-level metrics are constrained to particular language pairs, we selected En-Fr, En-De, and Zh-En as representative language pairs so that we could use BlonDe and contrastive evaluation sets. We chose to focus on the concatenation-based architecture given its simplicity, good performance, and widespread adoption in recent years [1] over alternative architectures, and we additionally explore different combinations of context window settings (1-2, 2-2, 3-1, etc.).
>
> [1] [Fernandes et al. Measuring and Increasing Context Usage in Context-Aware Machine Translation. Annual Meeting of the Association for Computational Linguistics. 2021](https://arxiv.org/abs/2105.03482) \
> \
> **2A.**
>
> “[In] L603, why do you say it shows difficulty to train on scratch on longer-sequence data?”
>
> We think that both reasons—the difficulty of training on longer-sequence data, as well as the relatively small size of the Para2Para corpus—could explain the extremely low performance of the NONE  (i.e., Transformer with no pre-training) setting (Table 7).
>
> To disentangle the two factors (is the dataset difficult because of its limited size, or because paragraph-to-paragraph translation is inherently a challenging task?), we also test additional baselines that leverage pre-training to alleviate the low-domain issue. Our pre-trained baselines (using the Transformer, LightConv, and mBART-25) all exhibit a marked performance improvement over the NONE setting, although the numbers across the board are relatively low, attesting to the difficult nature of paragraph-to-paragraph translation.
>
> We will clarify this point in our revision. \
> \
> **“Maybe you could prompt some LLMs [e.g., ChatGPT] to generate hypotheses on your test sets as well?”**
>
> How LLMs would fare on our Para2Para dataset is indeed an interesting question. From recent studies such as [1], we hypothesize that while LLMs can provide better paragraph-level translation (as they are better at handling long context), there would still be critical translation errors that a human translator would be able to avoid.
>
> As for using ChatGPT— given that OpenAI did not disclose the composition of ChatGPT’s training data, it is likely that there may be instances of data leakage from pre-training (especially as our dataset is sourced from public-domain data); thus, we do not believe these results represent a fair comparison with our other baselines in Table 7. The out-of-box ChatGPT results, included with the other pre-trained baselines on the Para2Para dataset, are as follows:
>
> | Domain | Pre-training   | BLEU  | BLONDE (all) | BLONDE (pron.) | BLONDE (entity) | BLONDE (tense) | BLONDE (d.m.) | COMET    |
> |--------|----------------|-------|--------------|----------------|-----------------|----------------|--------------|----------|
> | Closed | GPT-3.5     | 11.60 | 28.29        | **83.21**      | 35.63          | **73.98**      | **63.82**  | **0.7644** |
> |        | XFMR      | 15.80 | 34.80        | 78.55          | 50.39          | 71.87          | 60.19  | 0.7352   |
> |        | LightConv  | **16.80** | **36.86**    | 79.01          | **57.70**      | 72.90          | 61.18  | 0.7375   |
> |        | mBART25   | 15.70 | 35.60        | 78.80          | 55.06          | 72.05          | 58.92  | 0.7391   |
> | Open   | GPT-3.5    | 11.90 | **27.86**    | **86.02**      | 30.64          | **75.71**      | **66.93**  | **0.7648** |
> |        | XFMR       | 9.70  | 25.96        | 72.31          | **35.51**      | 65.71          | 51.43  | 0.7022   |
> |        | LightConv  | 8.90  | 24.89        | 73.41          | 30.31          | 68.14          | 52.87  | 0.7038   |
> |        | mBART25    | 8.10  | 23.06        | 72.09          | 23.37          | 67.39          | 51.23  | 0.7076   |
>
> [1] [Karpinska et al. Large language models effectively leverage document-level context for literary translation, but critical errors persist. arxiv preprint. 2023 ](https://arxiv.org/abs/2304.03245)

---

### Official Review · Reviewer_bWHY · 2023-08-04

**Soundness:** 4

**Excitement:**

3: Ambivalent: It has merits (e.g., it reports state-of-the-art results, the idea is nice), but there are key weaknesses (e.g., it describes incremental work), and it can significantly benefit from another round of revision. However, I won't object to accepting it if my co-reviewers champion it.

**Missing References:**

Zhang and Liu. Paragraph-Parallel based Neural Machine Translation Model
with Hierarchical Attention. Journal of Physics: Conference Series. 2020. https://iopscience.iop.org/article/10.1088/1742-6596/1453/1/012006/pdf
Note: The previous paper appears to be almost the same as Zhang et al. Paragraph-Level Hierarchical Neural Machine Translation. ICONIP. 2019.

[Contemporaneous] Ghussin et al. Exploring Paracrawl for Document-level Neural Machine Translation. EACL. 2023.

**Paper Topic And Main Contributions:**

This paper investigates challenges for context-aware neural machine translation. The authors show that discourse phenomena are sparse and that a context-aware transformer model (concatenating the context) doesn't perform well for some of them. They also replace the transformer model with another architecture (MEGA), with limited improvement, and argue for a better document-level translation metric. A paragraph-to-paragraph dataset of Chinese-English novels is also collected.

**Questions For The Authors:**

A. What does synthetic and analytic mean to describe languages?

B. For table 7, what would be the model performances before fine-tuning (i.e trained on WMT17 Zh-En only)?

**Reasons To Accept:**

The analysis, especially in sections 4.1 (discourse phenomena sparsity) and 4.2 (model performance for contextual phenomena), is fairly detailed. We can observe that some phenomena on which the context-aware models are struggling (e.g. tense, discourse markers) are infrequent (table 1).

The experiments are generally sound.

**Reasons To Reject:**

Section 4.3 ("The sentence-level NMT baseline is already competitive") is potentially misleading. Because BLEU scores are similar between context-aware and context-agnostic models, the authors mention that "This suggests that [...] context-agnostic models are already capable of delivering high-quality translations." However, this could also indicate that both types of model generate deficient translations.

High-levels observations in section 4 are often already known (although relevant papers are cited and the specific results/experiments are novel). Data sparsity is mentioned for example in [1]. Limited differences in BLEU scores have (in part) motivated the design of metrics to capture specific discourse phenomena (but these metrics are not perfect).

The collected paragraph-to-paragraph dataset is quite small, so its usefulness is not well established.

[1] Lupo et al. Divide and rule: Effective pre-training for context-aware multi-encoder translation models. ACL. 2022

**Reproducibility:**

4: Could mostly reproduce the results, but there may be some variation because of sample variance or minor variations in their interpretation of the protocol or method.

**Reviewer Confidence:**

4: Quite sure. I tried to check the important points carefully. It's unlikely, though conceivable, that I missed something that should affect my ratings.

**Typos Grammar Style And Presentation Improvements:**

Thai et al. Exploring document-level literary machine translation with parallel paragraphs from world literature. EMNLP. 2022. is duplicated in the references.

---

> ### Author Rebuttal · Authors · 2023-08-28
>
> We thank R1 for their time and feedback! We are encouraged that R1 finds that the analysis is “fairly detailed” and that the experiments are “generally sound,” and respond point-by-point to their comments below:
> \
> \
> **“Section 4.3….is potentially misleading….both types of model [could] generate deficient translations.”**
>
> We agree that the claim ("The sentence-level NMT baseline is already competitive") may be confusing.
>
> To be precise, we argue in Sect. 4.3 that the performance across the context-agnostic baseline and context-aware models is largely similar. Due to problems with common document-level datasets (e.g., relative lack of contextual signals) (Section 4.1),  context-aware models do not exhibit a meaningful improvement over context-agnostic models.
>
> Across different language pairs and model architectures, we find that this trend is consistent across not just sentence-level automatic metrics (e.g., BLEU, COMET), but surprisingly, across some document-level metrics as well (e.g., tense and discourse markers in BlonDe).
>
> Our empirical results could be interpreted to mean that both context-agnostic and context-aware models generate either high-quality or low-quality translations. The interpretation here is valid in either direction, and does not detract from our overall argument.
>
> As it is critical to keep our arguments precise, we will clarify the wording in our revision.
> \
> \
> **“High level observations in section 4 are often already known…Data sparsity is [previously] mentioned [and] limited differences in BLEU scores have…motivated the design of metrics to capture specific discourse phenomena.”**
>
> Some of the challenges in Section 4 have indeed been raised in prior work. We believe it is necessary to include them for the sake of completeness.
>
> However, in contrast to past literature, we conduct a more rigorous and finer-grained quantitative analysis of such observations over a range of language pairs, model architectures, concatenation schemas, and document-level phenomena. To date, no other work has attempted a comprehensive analysis of the obstacles that impede progress in context-aware NMT.
>
> For example, data sparsity is first posited by Lupo et al., 2022 [1], though they do not perform any data investigation and only focus on multi-encoder architectures. Our analysis is the first to quantitatively show the presence of context sparsity. By carefully examining document-level data (BWB, IWSLT), we statistically find that most sentences are short, simple constructions which do not require context to be accurately translated. We show that concatenation-based architectures also struggle with this phenomenon, resulting in meager document-level translation gains.
>
> As for the second point, it is well known that BLEU, COMET, etc. are inadequate measurements of document-level improvements, spurring the development of document-level metrics (e.g., BlonDe, contrastive evaluation sets) that explicitly  measure different discourse-level phenomena. Our study is the first to show that these metrics are only selectively helpful based on discourse category (Sect. 4.2). We additionally note several limitations in Sect. 4.5.
>
> [1]  [Lupo et al. Divide and rule: Effective pre-training for context-aware multi-encoder translation models. Annual Meeting of the Association for Computational Linguistics. 2022](https://arxiv.org/abs/2103.17151)
> \
> \
> **“The collected [Para2Para] dataset is quite small…its usefulness is not well established.”**
>
> At 10,545 paragraphs, our Para2Para dataset is indeed smaller than many benchmark sentence-level translation datasets, due to the challenge of finding paragraph-aligned, high-quality, professionally translated data. Para2Para is comparatively larger than the Zh-En split of Par3 [1], another open-sourced, paragraph-aligned literary translation dataset, which stands at only 1320 paragraphs.
>
> Although we are definitely open to expanding the dataset in future work, the main purpose of Para2Para is to establish preliminary baselines and demonstrate that taking the paragraph as a minimal discourse-level unit can serve as a challenging, realistic setting for context-aware NMT. We introduce pre-trained baselines to offset the impact of low domain size. From Table 7, these baselines perform noticeably better compared to a baseline training from scratch (the NONE setting), though the numbers across the board are all fairly low, showing that paragraph-to-paragraph translation is quite difficult in nature.
>
> [1] [ Thai et al. Exploring Document-Level Literary Machine Translation with Parallel Paragraphs from World Literature. Conference on Empirical Methods in Natural Language Processing. 2022.](https://arxiv.org/abs/2210.14250)
> \
> \
> **1A.**\
> In an analytic language, concepts can be mainly  expressed as root/stem words, and bound morphemes (affixes) are rare; examples include Chinese, Vietnamese, and Cambodian. In contrast, synthetic languages  have many affixes, in which multiple concepts would be fixed into a single word; examples include English, Latin, and French. The morpheme-to-word ratio is lower in analytic languages, and higher in synthetic ones. [1] We will add this elaboration in our revision.
>
> [1] O'Grady, W., Dobrovolsky, M., & Aronoff, M. (1997). Contemporary linguistics: an introduction. 2nd, U.S. ed.
> \
> \
> **1B.** \
> We include a table showing model performances without fine-tuning (i.e., trained only on WMT17 Zh-En) below:
> | Domain | Pre-training | BLEU  | BLONDE (all) | BLONDE (pron.) | BLONDE (entity) | BLONDE (tense) | BLONDE (d.m.) | COMET    |
> |--------|--------------|-------|--------------|----------------|-----------------|----------------|--------------|----------|
> | Closed | XFMR       | 7.30  | 25.42        | 67.41          | 37.24          | 60.82        | 54.06  | 0.6662   |
> |        | LightConv  | 6.30  | 23.30        | 58.19          | 34.32          | 56.16        | 47.57  | 0.6487   |
> |        | mBART25    | 6.70  | 21.35        | 63.56          | 22.10          | 60.96        | 47.52  | 0.6699   |
> | Open   | XFMR       | 4.70  | 21.22        | 59.81          | 35.45          | 54.31        | 44.12  | 0.6342   |
> |        | LightConv  | 4.20  | 20.78        | 53.35          | 38.94          | 49.80        | 39.96  | 0.6160   |
> |        | mBART25    | 4.10  | 18.61        | 55.89          | 26.27          | 54.39        | 39.35  | 0.6407   |
>
> For reference, this is the results including fine-tuning:
> | Domain | Pre-training   | BLEU  | BLONDE (all) | BLONDE (pron.) | BLONDE (entity) | BLONDE (tense) | BLONDE (d.m.) | COMET    |
> |--------|----------------|-------|--------------|----------------|-----------------|----------------|--------------|----------|
> | Closed | None       | 1.90  | 7.65         | 47.28          | 15.50           | 40.51          | 8.84   | 0.3725   |
> |        | XFMR      | 15.80 | 34.80        | 78.55          | 50.39           | 71.87          | 60.19  | 0.7352   |
> |        | LightConv  | **16.80** | **36.86**    | **79.01**      | **57.70**       | **72.90**       | **61.18**  | 0.7375   |
> |        | mBART25    | 15.70 | 35.60        | 78.80          | 55.06           | 72.05          | 58.92  | **0.7391** |
> | Open   | None      | 0.90  | 1.74         | 48.15          | 0.00            | 35.69          | 19.94  | 0.3429   |
> |        | XFMR      | **9.70**  | **25.96**    | 72.31          | **35.51**       | 65.71          | 51.43  | 0.7022   |
> |        | LightConv  | 8.90  | 24.89        | **73.41**      | 30.31          | **68.14**       | **52.87**  | 0.7038   |
> |        | mBART25    | 8.10  | 23.06        | 72.09          | 23.37          | 67.39          | 51.23  | **0.7076** |
>
> As expected, we can see that without additional training on the Para2Para dataset, all three models exhibit worse performance across the board.
> \
> \
> **Missing References** \
> We thank R1 for providing these two references and will add both to our revision! We should note that the first paper (Zhang and Liu, 2020) [1] is primarily a model paper.  While this work also introduces a parallel paragraph corpus, this dataset is divided based on word alignments, whereas our Para2Para dataset is aligned at the paragraph level. The second paper (Ghussin et al., 2023) [2] is technically concurrent work; the dataset in this paper is cultivated from Paracrawl, which sources from online webpages. Our dataset, in comparison, focuses exclusively on the literary domain.
>
> [1] [Zhang and Liu. Paragraph-Parallel based Neural Machine Translation Model with Hierarchical Attention. Journal of Physics: Conference Series. 2020.](https://iopscience.iop.org/article/10.1088/1742-6596/1453/1/012006) \
> [2] [Ghussin et al. Exploring Paracrawl for Document-level Neural Machine Translation. EACL. 2023.](https://arxiv.org/abs/2304.10216)

---

### Meta-Review · Area_Chair_mKVC · 2023-09-18

**Recommendation:** 4

**Metareview:**

This paper presents an extensive study of NMT systems that exploit context beyond the current sentence. It identifies several challenges to progress in this area, including sparsity of context signals and the need for accurate metrics. It proposes a simple framework based on paragraph-level translations, and presents baseline results on a new public dataset.

Reviewers were generally positive, finding the paper clearly written, and the analyses comprehensive and well organized, with solid experiments. The released dataset and benchmarks were also appreciated. On the negative side, one reviewer felt that the paper didn’t break new ground, and that the proposed paragraph-to-paragraph framework was overly simplistic.

Moving beyond sentence-level translation is an important problem for the field, and this paper goes further than previous studies in analyzing reasons for the relative lack of progress. Given the rather confused state of the art in this area, starting afresh with a clear and simple benchmark and dataset - armed with a detailed set of empirical observations about the signals available from extra-sentence context - seems like an excellent strategy. I think this work has the potential to make a large impact.

---

### Decision · Program_Chairs · 2023-10-07

**Decision:**

Accept-Main

**Comment:**

This paper presents an extensive study of NMT systems that exploit context beyond the current sentence. It identifies several challenges to progress in this area, including sparsity of context signals and the need for accurate metrics. It proposes a simple framework based on paragraph-level translations, and presents baseline results on a new public dataset.

Reviewers were generally positive, finding the paper clearly written, and the analyses comprehensive and well organized, with solid experiments. The released dataset and benchmarks were also appreciated. On the negative side, one reviewer felt that the paper didn’t break new ground, and that the proposed paragraph-to-paragraph framework was overly simplistic.

Moving beyond sentence-level translation is an important problem for the field, and this paper goes further than previous studies in analyzing reasons for the relative lack of progress. Given the rather confused state of the art in this area, starting afresh with a clear and simple benchmark and dataset - armed with a detailed set of empirical observations about the signals available from extra-sentence context - seems like an excellent strategy. I think this work has the potential to make a large impact.